# Spatial analysis of socioeconomic data and its relationship with illicit crops in Nariño-Colombia

**Andrés Fernando Grajales-Marín[1], Fabio Humberto Sepúlveda-Murillo[1]\*, Alex Tapia[1], Alexander Tabares[2]**

1 Faculty of Basic Sciences, University of Medellin, Medellin, Colombia, 2 Faculty of Economic and Administrative Sciences, University of Medellin, Medellin, Colombia

\* fhsepulveda@udemedellin.edu.co

## Abstract

The Sustainable Development Goals (SDGs) aim to eradicate poverty and promote sustainable development; however, socioeconomic disparities persist globally, particularly in Colombia. With a Gini index of 0.556 in 2022, Colombia ranks among the most unequal countries in Latin America, with its southwest region of Nariño facing severe socioeconomic challenges. Concurrently, Nariño registers the highest levels of coca cultivation in Colombia, accounting for 65% of national cocaine production, reflecting the region's precarious conditions. This study investigates the extent to which the spatial distribution of socioeconomic factors explains coca cultivation patterns in Nariño. Grounded in conflict economics, social capital, and social marginalization theories, the research constructs composite indices representing education, health, public services, economic conditions, and vulnerability. Using spatial analysis, it identifies areas with heightened poverty and vulnerability and examines their relationship with illicit crops. The findings highlight spatial non-stationarity in the factors influencing coca cultivation, offering region-specific insights and policy recommendations to combat illicit crops and foster sustainable development. These results provide a foundation for targeted interventions and contribute to broader strategies addressing inequality and illegal economies in Colombia.

**Data Availability Statement:** The data supporting the findings of this study are publicly available in PONE-Narino-2024 and can be accessed through

## 1. Introduction

In the contemporary global context, the Sustainable Development Goals (SDGs) have established themselves as an essential guide to address society's most pressing challenges. The SDGs, adopted by the United Nations, seek to eradicate poverty, protect the planet and ensure dignity and rights for all by 2030 [1]. However, despite global efforts, significant inequalities persist in the distribution of resources, opportunities and access to fundamental goods and services, perpetuating socioeconomic disparities around the world. These inequalities manifest themselves in a variety of ways, including income, education, health and employment, underscoring the urgent need to understand their causes in order to move towards a more equitable and sustainable society.

the following link: https://github.com/amtapia/PONE-Narino-2024.

**Funding:** The first author (Andres) received specific funding for this work from Ministerio de Ciencia, Tecnología e Innovación de Colombia. The funders had no role in study design, data collection and analysis, decision to publish, or preparation of the manuscript.

**Competing interests:** The authors have declared that no competing interests exist.

In Colombia, socioeconomic inequality remains a central challenge for the fulfillment of the SDGs. The country's Gini index reached a value of 0.556 in 2022, placing Colombia among the most unequal nations in Latin America [2]. In addition, various poverty indicators reveal the precariousness of the population's living conditions, such as the Monetary Poverty Index, which in 2022 affected 36.6% of Colombians, and the Multidimensional Poverty Index, which showed that 12.9% lacked the necessary conditions for individual development [3]. In this context, the state of Nariño, located in the southwest of the country, faces one of the most complex socioeconomic inequality situations in the region, characterized by limited economic development, especially in its agricultural sector, and a high level of unsatisfied basic needs [4,5].

Besides these socioeconomic disparities, Nariño faces an additional challenge related to the informal economy, specifically coca cultivation. According to the United Nations Office on Drugs and Crime report [6], the coastal regions of Nariño register the highest growth of illicit crops in Colombia, accounting for 65% of national cocaine production. This phenomenon reflects the precarious socioeconomic conditions in the region, which facilitates the expansion of illegal activities such as drug trafficking.

In this context, the central question of this research is: To what extent does the spatial distribution of socioeconomic conditions explain coca cultivation patterns in the state of Nariño? This question is posed because of the need to understand how socioeconomic conditions directly affect the prevalence of coca cultivation in the region, which could provide clues to develop more effective public policies aimed at eradicating illicit crops and promoting sustainable development alternatives.

The theoretical framework supporting this research is based on several fundamental theories that explain the socioeconomic factors associated with coca cultivation. The conflict economics theory suggests that armed conflict and violence have a direct impact on the economy and development, and in the case of Colombia, helps to understand how illegal armed groups control areas of coca cultivation, generating an environment conducive to drug trafficking [7,8]. Social capital theory highlights the importance of social networks and trust as key factors for economic and social development. In coca-growing communities, the lack of social capital hinders the implementation of crop substitution programs and the development of economic alternatives, lacking support networks [9,10]. Finally, social marginalization theory argues that social exclusion and lack of economic opportunities lead people to engage in illegal activities such as coca cultivation. In Colombia, the historical marginalization of rural communities and the lack of access to basic services such as education, health and employment contribute to perpetuate this phenomenon [11,12].

This study seeks to answer the research question by constructing composite indices that reflect key dimensions such as education, health, public services, economy and vulnerability in Nariño. Through a spatial analysis of these socioeconomic conditions, it aims to identify areas with higher rates of poverty and vulnerability and examine how these directly affect the prevalence of illicit crops. In addition, the study will model the spatial non-stationarity of the factors associated with coca cultivation and provide detailed results on the most relevant factors in each area of Nariño, which will allow the formulation of specific recommendations for public policies.

The structure of the article is as follows: Section 2 describes the methodology employed in the research; Section 3 presents the results and discussion of the findings; and Section 4 concludes with a discussion of the implications of the results and recommendations for future research and for government decision-making.

## 2. Methodology

### 2.1 Area of study, variables, and information sources

This investigation was conducted in the state of Nariño, located in the southwestern region of the Republic of Colombia. Encompassing an area of 33,268 km², the state, as per recent data from DANE, is inhabited by a total population of 1,627,589 individuals. The administrative territorial structure comprises 64 municipalities, which constitute the units under scrutiny in this study. The Municipality of Pasto serves as the capital of the state (Fig 1).

Nariño, as a state, exists within the dichotomy of being strategically positioned for Colombia owing to its geographical location, agricultural potential, and prospective industrial development, for example, [13] in which the potential of the Nariño in terms of biodiversity and national and international connectivity is presented. Simultaneously, it unfortunately garners recognition in regional and international contexts for issues related to drugs and violence, as evidenced by reports and studies conducted by various offices of multilateral organizations [14–16]. These reports, particularly those from national entities [17], acknowledge its potential as a special border zone due to its proximity to Ecuador in the south and its possession of the port of Tumaco, connecting it to the Pacific Ocean in the northwest. However, the region currently attains global recognition primarily due to illegal activities that have detrimental effects on people's well-being, contributing to a pervasive stigma across various levels. Moreover, several municipalities within the state of Nariño contend with highly intricate conditions of economic and social marginalization.

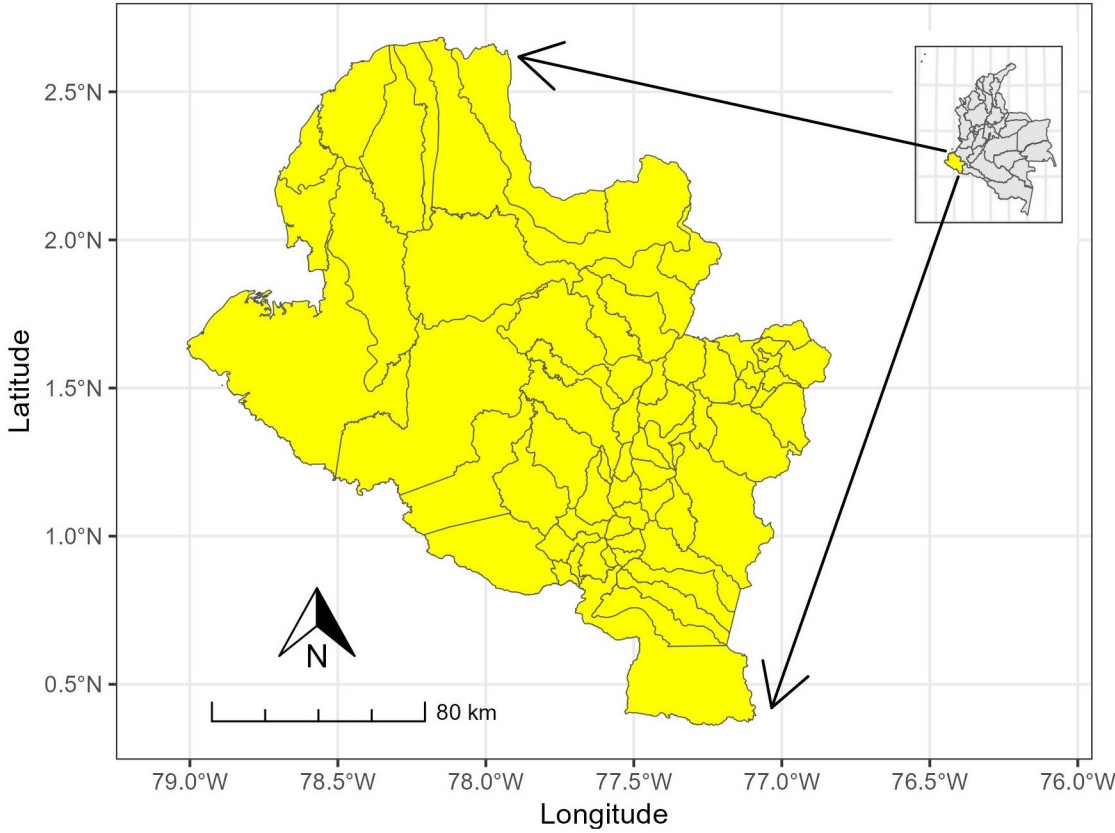

**Fig 1. Location of the state of Nariño in Colombia (inserted small map).**

To measure the economic and social development of the state of Nariño, six composite indices were meticulously constructed for each of its 64 municipalities. Each index integrates multiple quantitative variables corresponding to the unique characteristics of each municipality. The Educational Performance Index (EPI) captures information from the results of a national test (called Prueba Saber 11) in Mathematics and Spanish Language. The variables related to education coverage in both transitional and secondary education contributed to the construction of the Education Coverage Index (ECI). The Health Coverage Index (HCI) was formulated using data on the population affiliated with one of Colombia's health regimes and the demographic segment under one year of age that received the third dose of the pentavalent vaccine. The Public Services Coverage Index (PSCI) was crafted by considering critical services such as electricity, internet, aqueduct, and sewerage coverage. The Economic Index (EI) is contingent upon metrics related to GDP per capita and the employed population of the municipality. To construct the Vulnerability Index (VI), parameters such as overcrowding, the population in a state of misery, and instances of child labor were systematically integrated.

Furthermore, three pivotal variables integral to the dynamics of the state's economy were taken into consideration. Firstly, an examination of the homicide rate (HOMI) in each municipality was conducted. Secondly, an evaluation of the connectivity and infrastructure within the state was undertaken, with a specific focus on the variable 'Roads'. This variable was systematically constructed by factoring in the length of primary roads in each municipality. Lastly, an assessment of the production of cocaine cultivation in each of the state's municipalities was included. The homicide rate, connectivity variables, along with the meticulously constructed indices, served as the independent variables in the initial phase to analyze their correlation with cocaine cultivation as the response variable.

Data for this study were provided from various sources of information. Specifically, the socioeconomic variables for the state of Nariño were extracted from the DANE database. Information related to cocaine production was provided by the Ministry of Justice and Law. Data concerning the length of primary roads in each of the state's municipalities were obtained from OpenStreetMap (OSM) [18], a collaborative project for the creation of open-access maps. Table 1 provides a detailed overview of the six indices, their associated variables, and sources of information. It is important to note that the databases originate from various sources, which means the data were collected in different years due to the country's policies.

## 2.2 Statistical analysis

Given the disparate origins of the data, an initial merge was executed, consolidating all information into a unified spatial database that encompasses records for each municipality. The statistical analysis of the dataset involved the computation of summary statistics, providing a comprehensive overview of both the variables employed in the study and the constructed indices. Additionally, traditional, and spatial statistical graphs, some of which are presented in this paper, were employed to analyze the global and local dynamics of variables and indices.

To elucidate the dynamics of cocaine production in the state of Nariño with respect to the indices, homicide rate, and roads, an econometric model was systematically formulated. In the following subsections, we present a succinct overview of the theory guiding index construction and a brief description of the econometric model designed to explore their relationship with cocaine production.

**2.1.1 Constructing composite indices.** Initially, 6 synthetic composite indices were estimated for the 64 municipalities of the state of Nariño. The procedure used to construct the indices was based on the Distance-Learning (DL2) proposed by [19].

**Table 1. Description of the variables considered to construct the indices.**

| Index / Abbreviation | Variable / Abbreviation | Description | Source of Information / Year |
|---|---|---|---|
| Educational Performance Index/ EPI | Mathematics / Math | Average score in the mathematics module of the national test (Prueba Saber 11) based on students residing in the municipality. | Colombian Institute for the Evaluation of the Quality of Education (ICFES) / 2021 |
| | Language / Lang | Average score in the Spanish language module of the national test (Prueba Saber 11) based on students residing in the municipality. | Colombian Institute for the Evaluation of the Quality of Education (ICFES) / 2021 |
| Educational Coverage Index/ ECI | Transition / Tran | The ratio between the number of students enrolled in an educational level who are theoretically old enough to attend it and the total population corresponding to the same age. | Ministry of Education (Integrated Enrollment System–SIMAT) / 2021 |
| | Media / Me | The ratio between the number of students enrolled in an educational level who are theoretically old enough to attend it and the total population corresponding to the same age. | Ministry of Education (Integrated Enrollment System–SIMAT) / 2021 |
| Health Coverage Index/ HCI | Health / Salt | Proportion of the population enrolled in one of the health systems | Ministry of Health / 2021 |
| | Pentavalent / Pent | Proportion of the population under 1 year of age that has received the third dose of pentavalent vaccine. | Ministry of Health / 2020 |
| Public Service Coverage Index / PSCI | Rural electric power / EnerRural | Percentage of electricity coverage in rural areas. | Mining and Energy Planning Unit (UPME) / 2019 |
| | Internet / Inter | Percentage of subscribers with dedicated Internet access out of total population | Ministry of Information and Communications Technologies (MINTIC) / 2021 |
| | Aqueduct / Acue | Percentage of coverage of the water service reported by the territorial entities in the Stratification and Coverage Report. | Superintendencia de Servicios Públicos Domiciliarios (Stratification and Coverage Report–REC) / 2021 |
| | Sewerage / Water Supply | Percentage of coverage of the Sewerage service reported by the territorial entities in the Stratification and Coverage Report. | Superintendencia de Servicios Públicos Domiciliarios (Stratification and Coverage Report–REC) / 2021 |
| Economic Index / EI | Value added per capita / GDP | Measures GDP per capita | DANE/ 2020 |
| | Employment / Emp | Percentage of people formally employed as a percentage of total population | DANE/ 2016 |
| Vulnerability Index / VI | Overcrowding / Hacin | Percentage of households facing housing resource deprivation | DANE/ 2020 |
| | Misery / Mise | Percentage of people in each municipality living in extreme poverty | DANE/ 2018 |
| | Child Labor / TraInf | Percentage of households with at least one child between 12 and 17 years of age working | DANE/ 2020 |
| | Homicide / HOMI | Ratio of homicide cases per 10,000 inhabitants | Ministry of Defense / 2021 |
| | Roads / Road | Proportion of primary roads in each municipality | OpenStreetMap (OSM) / 2023 |
| | Cocaine / Coca | Proportion of land under cocaine cultivation in each municipality | Ministry of Justice and Law / 2021 |

Let $X$ denote a matrix of size $m \times n$, where the $m$ columns represent the quantitative variables and the $n$ rows represent the observations or spatial units (municipalities, countries, regions, etc.). Initially the variables are normalized by a change of scale. Subsequently, let $Z$ denote the matrix of size $m \times n$ containing the standardized variables. Then, the DL2 is defined as follows:

$$DL2(Z_s,\ Z_t) = \left( \sum_{j=1}^{m} |\mathcal{Z}_{sj} - \mathcal{Z}_{tj}|^2 \omega_j \right)^{1/2} \tag{1}$$

where $s$ and $t$ are two compared units or observations and $\omega_j$ are the weights that are calculated using iterative machine learning algorithms. This definition (function) considers the concept of proximity between units, allowing comparisons to be made between the spatial units (in our case municipalities) studied and territorial disparities to be identified. According to its construction, the values taken by all indices are between 0 and 1.

Once the six indices were calculated for all municipalities, they were combined (summed) to determine a single Multidimensional Index (MI) for each municipality $j$. The MI was calculated as follows [20].

$$(\text{MI})_j = (\text{EP}I)_j + (\text{ECI})_j + (\text{HCI})_j + (\text{PSCI})_j + (\text{EI})_j + (\text{VI})_j \tag{2}$$

Finally, to better interpret the results, the MI is scaled between 0 and 1 using the following expression:

$$(\text{MI}_{scale})_j = \frac{(\text{MI})_j - \min(\text{MI})_j}{max(\text{MI})_j - min(\text{MI})_j} \tag{3}$$

A value close to 0 means low living conditions of the population and low economic growth, and a value close to 1 indicates good living conditions of the population and high economic growth of the municipalities of the state of Nariño. Both the descriptive analysis and the spatial distribution of each index in the study area were visualized and analyzed.

**2.1.2 Econometric models.** This research employs global and local regression models to explore the relationship between the proportion of hectares dedicated to cocaine cultivation (response variable) and the indices, homicide rate and road (explanatory variables) in the state of Nariño. To compare and select significant independent variables, initially a global regression was used followed by a local extension called geographically weighted regression (GWR). The latter method facilitated the examination of spatial heterogeneity of the relationship between the response variable and the explanatory variables.

In the examination of the relationship between a response variable $Y$, and a set of independent variables $X_1, X_2, \ldots, X_p$, the analytical framework involves an Ordinary Linear Regression (OLR) model:

$$y_i = \beta_0 + \sum_{k=1}^{p} \beta_k x_{ik} + \varepsilon_i \tag{4}$$

where $\beta_0, \beta_1, \ldots, \beta_p$ are the parameters and $\varepsilon_1, \varepsilon_2, \ldots, \varepsilon_n$ are the error terms. In this global model, the estimated coefficients $\beta_k$ are considered constant throughout the study area. However, the hypothesis of spatial uniformity of the effect of the explanatory variables on the dependent variable is often unrealistic [21]. Then, to consider the geographical non-stationarity of the relationship and incorporate the spatial structure, an extension of the model represented by the Eq (1), referred to as GWR, is introduced. This extension involves the estimation of local parameters for each geographic location in the dataset, as defined by [22]:

$$y_i = \beta_0(u_i,\ v_i) + \sum_{k=1}^{p} \beta_k(u_i,\ v_i) x_{ik} + \varepsilon_i \tag{5}$$

where $(u_i, v_i)$ denotes the geographic coordinates at location $i$ (in this study, these were the coordinates of the centroids of each of the municipalities of Nariño), $y_i$ is the value of the dependent variable at location $i$; $x_{ik}$ is the value of the kth independent variable at location $i$; $p$ is the number of independent variables; $\beta_{ik}$ is the local regression coefficient for the kth independent variable at location $i$; and $\varepsilon_i$ is the random error at location $i$. In the calibration of Eq (5), it is implicitly assumed that data observed near the location have more influence on the estimation than data located farther away from. Next, the model measures the inherent relationships around each regression point $i$, where each set of regression coefficients is estimated using a weighted least squares approach. Thus, the matrix expression for this estimation is given by [23]:

$$\hat{\boldsymbol{\beta}}(u_i, v_i) = [\boldsymbol{X}^T \boldsymbol{W}(u_i, v_i)\boldsymbol{X}]^{-1}\boldsymbol{X}^T \boldsymbol{W}(u_i, v_i)\boldsymbol{y} \tag{6}$$

where $X$ is the matrix of predictor variables with a column of 1s for the intercept, $y$ is the vector of response variable, and $W(u_i,v_i)$ is a weighting matrix of size $n \times n$ whose off-diagonal elements are zero and whose diagonal elements denote the geographic weighting of each of the $n$ observed data for regression point $i$ at location $(u_i,v_i)$. There are three key elements in the construction of this weighting matrix: (i) the type of distance, (ii) the kernel function, and (iii) its bandwidth. For this work considering the irregular topography of the state of Nariño, the Euclidean distance, the Bi-squared function and an adaptive bandwidth were used [24–26].

Since extreme values can generate biased results, the GWR was performed with a robust analysis to mitigate this issue [24]. Furthermore, for constructing the spatial database, estimating, and mapping various measurements, R software was employed [27].

## 3. Results and discussion

### 3.1 Descriptive analysis and spatial distribution

Table 2 provides an overview of the behavior of the response variable (Coca) and the independent variables (Indices, HOMI, and Roads) through eight global descriptive statistics:

minimum (Min), quartile 1 (Q1), median (Median), mean (Mean), quartile 3 (Q3), maximum (Max), coefficient of variation (CV), and coefficient of asymmetry (CA). Fig 2 displays the statistical distribution of the indices using box-and-whisker plots.

The EPI, HCI, PSCI and VI indices exhibit values relatively close to 1. According to the Q1 of these indices, 75% of the municipalities have a high mean value. These results indicate that municipalities have a good educational performance, with good coverage in education, public services, and health. Along with low vulnerability. However, these indices display left-skewed distribution with the presence of outliers (CA), signifying the existence of municipalities with lower values and, consequently, indicating areas with inadequate protection.

The EI is characterized by values close to 0, with most municipalities not surpassing a value of 0.1585 (Q3). This indicates a generally low economic performance across the municipalities. However, the corresponding box-and-whisker diagram exhibits an asymmetric trend towards the right side with extreme values (as indicated by CA value). This asymmetry reflects a few municipalities with notably higher economic performance in the state of Nariño. Furthermore, based on the CV values, all indices and variables display a high degree of dispersion. The descriptive metrics and the statistical distribution show the existence of significant developmental differences (educational, social, economic, health, etc.) among the municipalities under investigation.

**Table 2. Global descriptive statistics of variables and constructed indices.**

| Response Variable | Min | Q1 | Median | Mean | Q3 | Max | CV (%) | CA |
|---|---|---|---|---|---|---|---|---|
| Coca | 0,0000 | 0,0000 | 0,0000 | 0,0100 | 0,0085 | 0,0600 | 178,5000 | 2,0700 |
| Explanatory Variables | | | | | | | | |
| EPI | 0,0312 | 0,6031 | 0,7035 | 0,6400 | 0,7759 | 0,9605 | 33,0800 | -1,2700 |
| ECI | 0,0015 | 0,3992 | 0,5356 | 0,5000 | 0,6058 | 0,9501 | 38,6700 | -0,3800 |
| HCI | 0,3155 | 0,6490 | 0,7503 | 0,7300 | 0,8431 | 1,0000 | 19,5600 | -0,5600 |
| PSCI | 0,0736 | 0,6235 | 0,6845 | 0,6700 | 0,8008 | 0,8927 | 24,1300 | -1,2500 |
| EI | 0,0000 | 0,0981 | 0,1417 | 0,1600 | 0,1585 | 0,7592 | 85,9500 | 2,9600 |
| VI | 0,1710 | 0,3141 | 0,3958 | 0,4300 | 0,5177 | 0,9343 | 41,4200 | 0,8500 |
| HOMI | 0,0000 | 0,6225 | 1,7250 | 3,5589 | 5,3750 | 18,8100 | 122,1316 | 1,7303 |
| Roads | 0,0000 | 0,0000 | 0,0000 | 0,2200 | 0,3654 | 1,3122 | 144,3900 | 1,4700 |

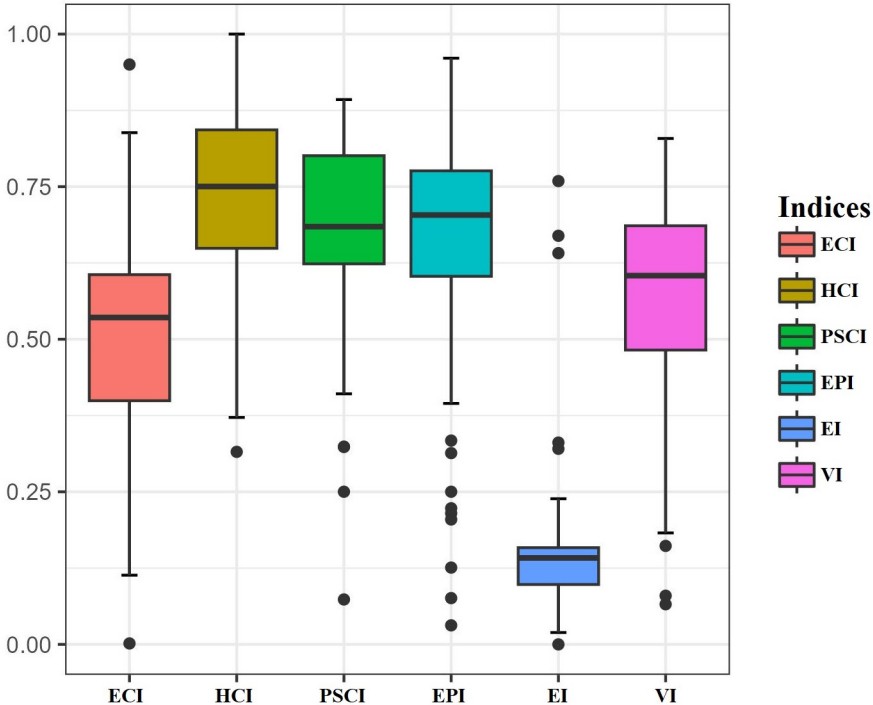

**Fig 2. Statistical distribution of the indices.**

The spatial distribution of the proportion of hectares of cultivated cocaine and the location of primary roads in the state of Nariño is depicted in Fig 3. This distribution reveals the geographic variability of cocaine production in the state, showing a concentration of high values in the northwestern part of the state of Nariño. This concentration can be attributed to the significant pressure exerted by non-government armed groups aiming to control several municipalities in this area [14]. Additionally, the proximity of these municipalities to the Pacific Ocean facilitates illegal exportation to other countries.

In contrast, municipalities with lower cocaine production tend to be situated in the southeast, closer to Pasto city, the capital of Nariño. Furthermore, a notable observation is the poor road infrastructure in most municipalities of the state, with many of these coinciding with areas of high cocaine production. This observation aligns with findings from other studies indicating that municipalities with limited roads, infrastructure, connectivity, and access tend to experience an increase in illicit crops [28–30].

Fig 4 illustrates the spatial distribution of the constructed indices and the homicide rate, revealing the spatial non-stationarity of these characteristics in the state of Nariño. The distributions of the indices EPI, ECI, HCI, PSCI, VI, and the HOMI variable exhibit a consistent spatial pattern from northwest to southeast, dividing the state into two segments. The southeastern part comprises municipalities with favorable living conditions and economic development, while the northwestern part faces social and economic fragility. The EI index demonstrates a distinctive spatial distribution, indicating a prevalence of municipalities with low economic performance, except for Potosi and Pasto, the capital of Nariño. The spatial distribution of MI (Fig 3h) summarizes the behavior of these indices, reinforcing the observed division and emphasizing the socioeconomic inequality prevalent in the state of Nariño.

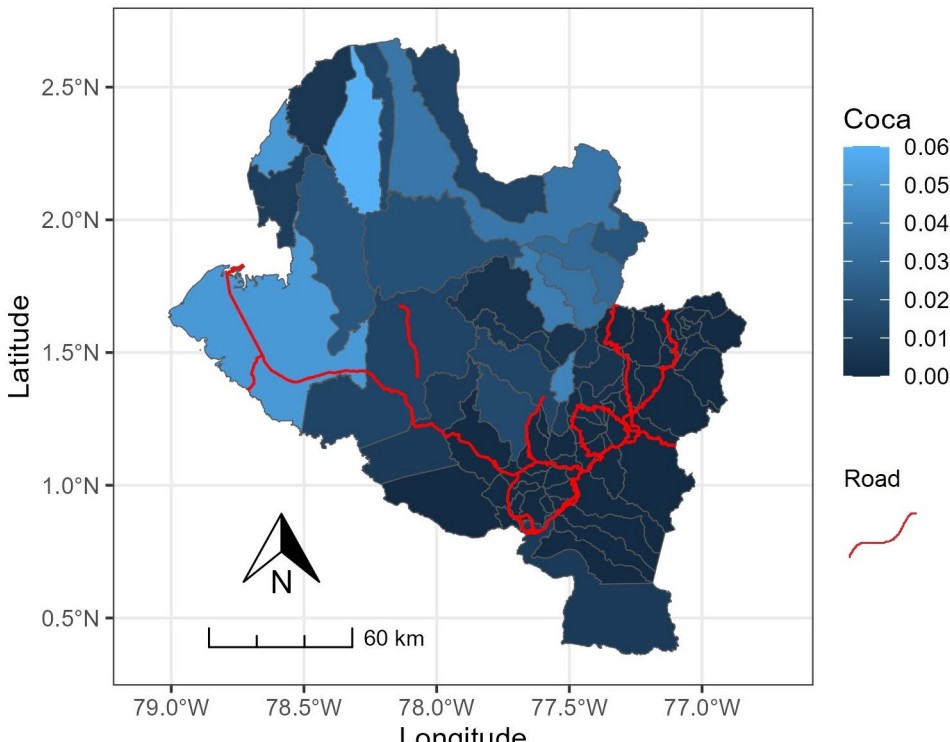

**Fig 3. Spatial distribution of cocaine cultivation (in blue) associated with main roads and trunk roads (red lines).**

## 3.2 Spatial econometric model

Initially, a global regression model (Ordinary Least Square) as a benchmark, performance comparison and attribute selection were applied. Table 3 shows the results of the global and GWR models: significant estimated coefficients, variance inflation (VIF), the $R^2$, adjusted $R^2$ and residual sum of squares. According to these results, two variables were significant in explaining cocaine production in the state of Nariño: EPI and HOMI. Collinearity was tested by analyzing the VIF, with values below the common threshold, indicating no multicollinearity issue.

The estimated coefficients in both global and local models were consistent with their expected signs. Assessing goodness-of-fit measures, R2 and adjusted R2 in the GWR notably improved from 0.468 and 0.450 to 0.647 and 0.539, respectively. The analysis of residuals (sum of squares residuals) indicated a superior fit in the GWR. Moreover, the examination of spatial variation in the explanatory power of the model revealed notable improvements (spatial distribution not presented in the paper). The comparative results underscored that the explanatory power of the local regression model was significantly higher than that of the global regression model, which is consistent with the results of the reviewed studies.

Undoubtedly, the most important results of the modeling reside in the estimated local coefficients of each of the explanatory variables. These coefficients provide a means to comprehend the spatial variation visually and analytically in the influence of each variable on cocaine production. The spatial distribution of the parameter estimates, and their significance are shown in Fig 5. According to these spatial distributions and based on the descriptive statistics (Min, Median, Max) of the local estimates of the GWR coefficients (Table 3), there are significant

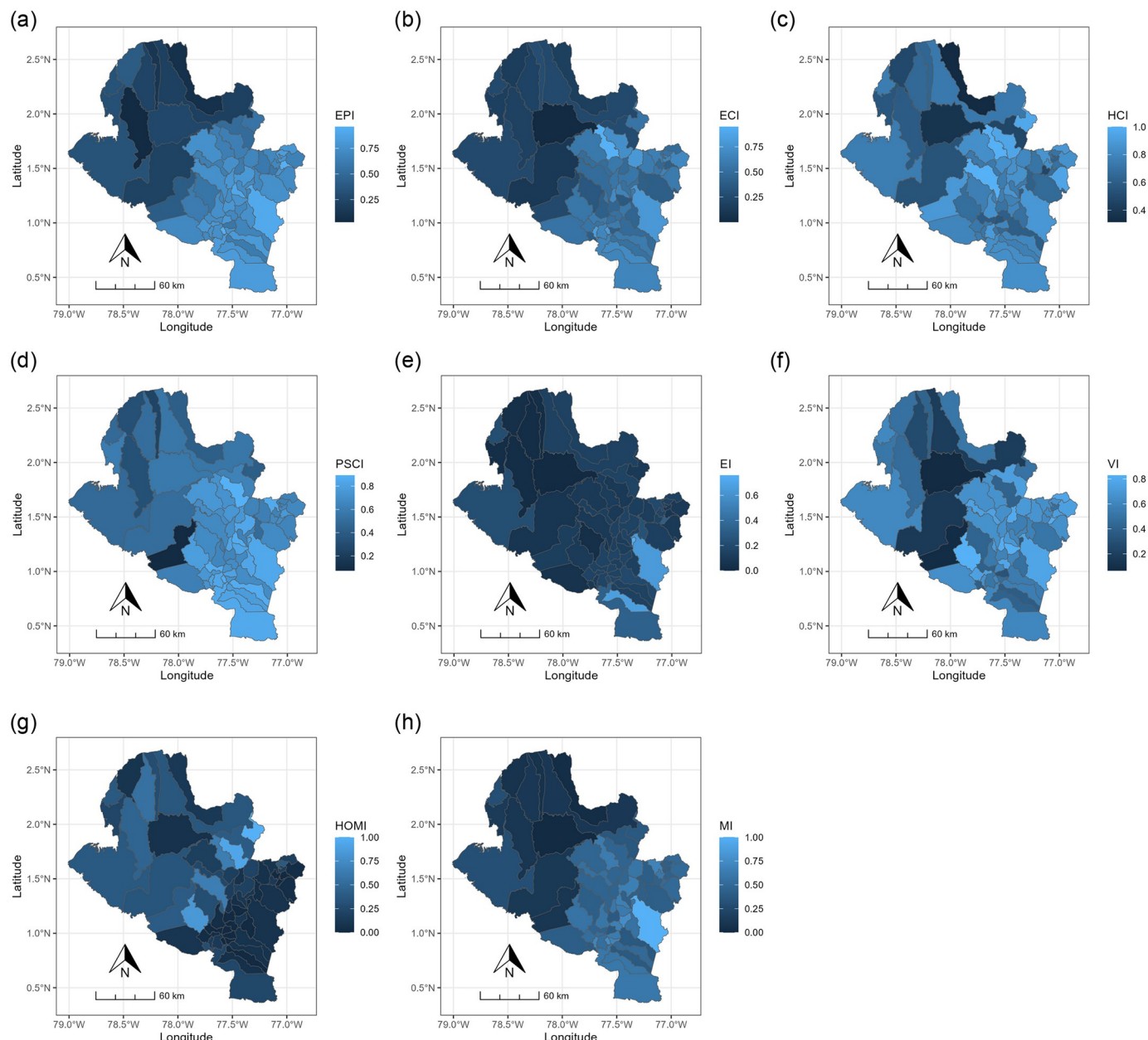

**Fig 4. Spatial distribution of the indices and HOMI.** The color bar to the right of the maps shows the values of each one.

variations in the relationships between the two independent variables (EPI and HOMI) and cocaine production in the State of Nariño.

The negative sign of the estimated coefficients on the EPI variable (Fig 4a) is expected and it indicates that there is an inverse relationship between EPI and cocaine production (although a change in the sign of the EPI coefficients is observable, it lacks significance). This correlation is consistent with findings in existing literature, indicating that cocaine crop production tends to decrease with higher levels of education [28,31–33]. Notably, this relationship is not uniform across all municipalities (Fig 5a), and the effect of EPI is more pronounced and

**Table 3. Global and local model results.**

| Response Variable: Coca | | | | | |
|---|---|---|---|---|---|
| Explanatory Variables | OLS coefficient estimates | Summary of GWR coefficient estimates | | | VIF |
| | | Min | Median | Max | |
| EPI | - 0.025 ** | - 0.031 | - 0.003 | 0.022 | 1.130 |
| HOMI | 0.026 *** | 0.002 | 0.023 | 0.048 | 1.130 |
| Model statistics | | | | | |
| R2 | 0.468 | 0.647 | | | |
| Adjusted R2 | 0.450 | 0.539 | | | |
| Residual sum of squares | 0.006 | 0.004 | | | |

Notes:

**p<0.001,

***p<0.0001;

VIF = Variance Inflation Factors.

statistically significant in a considerable number of municipalities in the state of Nariño (Fig 5b). The region where this relationship holds significance corresponds to municipalities exhibiting low educational performance (Fig 4a), coinciding with the area of high cocaine production (Fig 3) and representing the less developed part of the state, as illustrated by the spatial distribution of the MI (Fig 4h). This pattern can be attributed to a demand that surpasses the educational resources in this zone, characterized by difficult access (distant from the capital city of Nariño). Consequently, some municipalities lack the necessary infrastructure for providing educational services and often face a shortage of qualified educators, leaving numerous young individuals susceptible to engaging in activities such as cocaine production.

Finally, Fig 5c reveals a direct and spatially non-uniform correlation between the homicide rate and cocaine production. This finding aligns with the outcomes of previous studies that highlight a positive connection between increased illicit crop cultivation in regions marked by security issues and a limited government presence [31,34–39]. Specifically in Colombia, the evidence indicates that, on average, the homicide rate tends to rise in municipalities with cocaine cultivation since 2015 [36,40,41].

Nevertheless, the impact of the homicide rate on cocaine production does not demonstrate significance uniformly across all municipalities in the state of Nariño. Fig 5d delineates the municipalities where the association is statistically significant at a 95% confidence level. Notably, it is evident that municipalities primarily situated in the northwest of the state exhibit a substantial and statistically significant correlation between the homicide rate and cocaine cultivation. This region corresponds to municipalities characterized by elevated homicide rates (Fig 4g) aligning with the areas of heightened cocaine production (Fig 3). Thus, the findings of this study corroborate the assertions of the referenced authors, affirming that municipalities with elevated homicide rates also exhibit increased cocaine cultivation.

## 4. Conclusions

The SDGs serve as a crucial framework for tackling global imbalances and striving toward a more inclusive and sustainable future. However, entrenched socioeconomic disparities pose a substantial challenge to realizing these objectives. To inform the design and implementation of comprehensive solutions addressing the root causes of inequality and fostering sustainable and just development, this study quantified and spatially analyzed the socioeconomic and

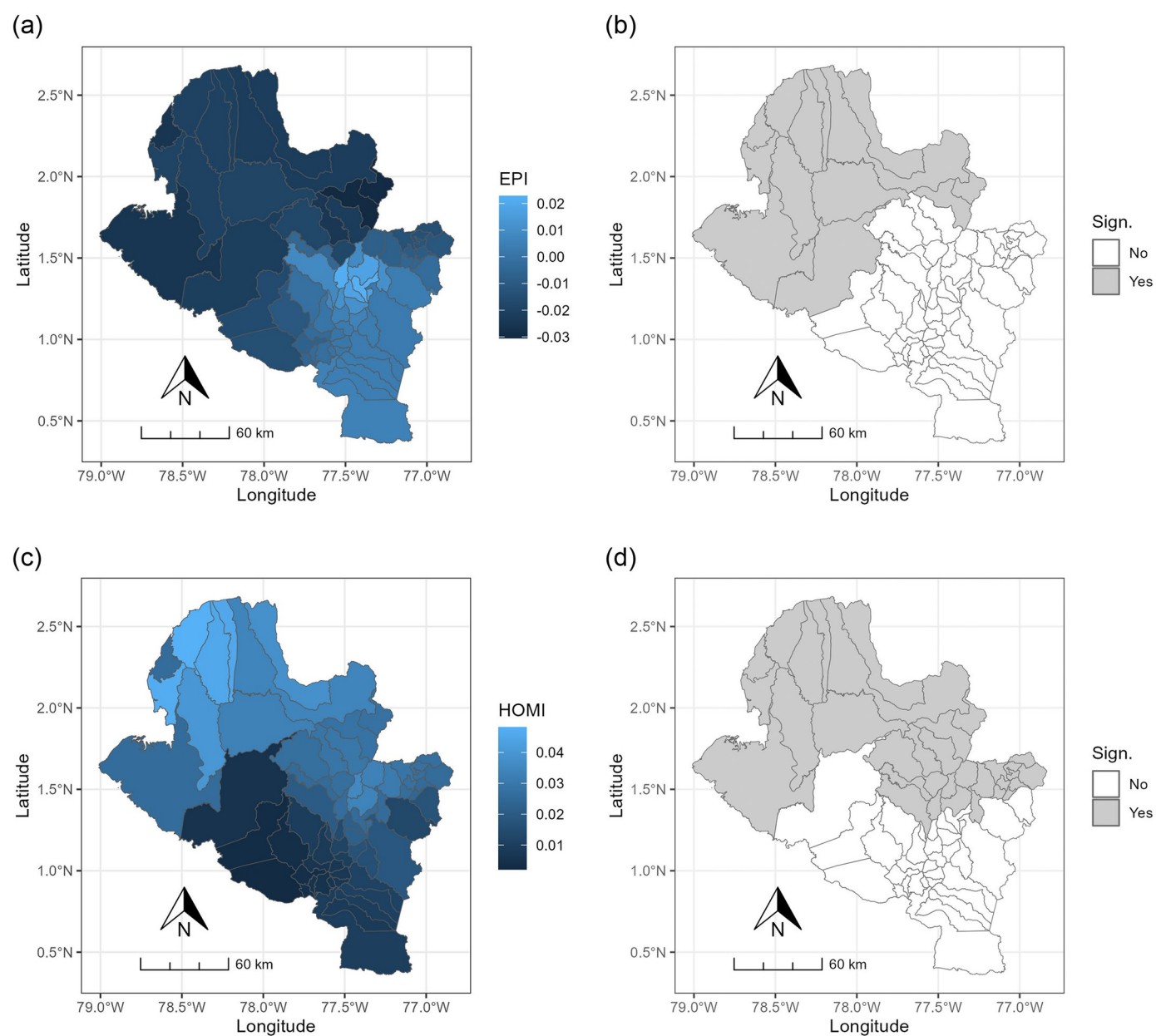

**Fig 5. Spatial distribution of GWR coefficients and their significances.**

territorial conditions of municipalities in the state of Nariño, one of the most unequal and least developed states in Colombia. The initial segment of this article employed statistical methodologies to construct diverse composite indices. Subsequently, in the latter part of the study, an econometric analysis was conducted utilizing these indices and additional variables, such as the homicide rate and road infrastructure to elucidate their potential role in explaining cocaine production across the municipalities.

The DL2 was used to create the indices. These indices were combined to build a single index called MI, serving as a comprehensive summary that encapsulates the information encompassed by the individual indices. The spatial distributions of all the indices reveal a

discernible spatial heterogeneity in social and economic inequality. This spatial diversity substantiates the existing disparities in opportunities among municipalities, indicating that their populations do not experience optimal conditions and lack basic capabilities such as education, health, and public services, as per the framework outlined by [42,43]. The observed spatial pattern distinctly divides the state into two zones: one exhibiting favorable social and economic conditions (Southeast) and another manifesting suboptimal performance (Northwest). The delineated spatial heterogeneity provides valuable insights for the formulation of targeted public policies and the implementation of programs aimed at enhancing the overall quality of life for the populace.

The econometric analysis employed both global and local regression methodologies. Initial application of global regression served a dual purpose: variable selection based on significance and as a benchmark for comparison. Two variables, EPI and HOMI, emerged as statistically significant. Subsequently, GWR was employed to delve into the spatial nuances of the relationship between cocaine production and the inherent characteristics of each municipality. The findings indicated that GWR outperformed global regression in explaining cocaine production in the state of Nariño. This suggests that all estimated parameters exhibit a discernible spatial variation pattern, enabling nuanced conclusions specific to each municipality.

The anticipated type of dependence between cocaine production and the two explanatory variables, EPI and HOMI, manifested as expected: negative for EPI and positive for HOMI. However, the impact of these variables on cocaine production did not achieve significance across all spatial units within the study area. GWR mapping results illustrated municipalities in the northwestern part of the state exhibiting a noteworthy correlation between cocaine cultivation issues and education as well as homicide cases. These outcomes align with the findings of the [16], highlighting a consistent rise in cocaine cultivation in this region over the past decade. The report identifies various factors contributing to this increase, some of which are related to EPI and HOMI, including the heightened global demand for cocaine, expectations stemming from peace agreements, an increase in illegal drug trafficking actors, the persistence of territorial vulnerability, and heightened incentives for cocaine production.

This study provides evidence suggesting a close relationship between socioeconomic conditions and the spatial distribution of coca cultivation in the state of Nariño, Colombia. The results support the social marginalization theory by showing that areas with high poverty and exclusion have a higher concentration of illicit crops, suggesting that the lack of access to basic services and economic opportunities creates a favorable environment for the informal economy and activities related to drug trafficking. This finding is consistent with social capital theory, as it indicates that the lack of social cohesion in these communities could limit the effectiveness of development programs that seek to replace coca cultivation. Based on the theory of conflict economics, the results also indicate that socioeconomic factors in Nariño not only influence coca cultivation but are also related to conflict dynamics in areas where coca cultivation is controlled by illegal armed groups. This conflict-prone environment, in combination with social marginalization and lack of social capital, contributes to the perpetuation of dependence on coca cultivation as the main livelihood in certain areas.

In addition, the results suggest that illicit crop reduction policies must be adapted to the specific conditions of each area and cannot be uniform. Intervention in Nariño requires a comprehensive approach that addresses socioeconomic conditions and improves community cohesion to facilitate viable and sustainable economic alternatives. Limitations of this study include the lack of detailed local data on social capital and armed group activity, which could enrich the analysis in future research.

In conclusion, this study helps to understand how socioeconomic conditions affect the distribution of coca cultivation in Nariño, underscoring the importance of specific interventions

that consider the particular social and economic dynamics of the region. Public policies should be designed in a way that not only addresses socioeconomic conditions, but also promotes the strengthening of social capital and the reduction of armed conflict, creating an environment more conducive to sustainable development and peace.

## Author Contributions

**Conceptualization:** Andrés Fernando Grajales-Marín, Fabio Humberto Sepúlveda-Murillo, Alex Tapia.

**Data curation:** Andrés Fernando Grajales-Marín, Fabio Humberto Sepúlveda-Murillo, Alex Tapia.

**Formal analysis:** Andrés Fernando Grajales-Marín, Fabio Humberto Sepúlveda-Murillo, Alex Tapia, Alexander Tabares.

**Funding acquisition:** Andrés Fernando Grajales-Marín.

**Investigation:** Fabio Humberto Sepúlveda-Murillo, Alex Tapia.

**Methodology:** Andrés Fernando Grajales-Marín, Fabio Humberto Sepúlveda-Murillo, Alex Tapia, Alexander Tabares.

**Validation:** Andrés Fernando Grajales-Marín, Fabio Humberto Sepúlveda-Murillo, Alex Tapia.

**Visualization:** Alex Tapia.

**Writing – original draft:** Alexander Tabares.

**Writing – review & editing:** Andrés Fernando Grajales-Marín, Fabio Humberto Sepúlveda-Murillo, Alex Tapia.

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
