## [Decision Letter · Decision Letter 0]

16 Oct 2024

PONE-D-24-24934Spatial Analysis of Socioeconomic Data and its Relationship with Illicit Crops in Nariño-ColombiaPLOS ONE

Dear Dr. Sepúlveda-Murillo,

Thank you for submitting your manuscript to PLOS ONE. After careful consideration, we feel that it has merit but does not fully meet PLOS ONE’s publication criteria as it currently stands. Therefore, we invite you to submit a revised version of the manuscript that addresses the points raised during the review process.

**The reviewers have opposing opinion on the merits of the paper. On the one hand, reviewer 1 suggests that the paper provides valuable insights into the complex relationship between socioeconomic factors and cocaine cultivation, and its findings have important implications for policy-making in Colombia. I do agree with reviewer 1´s appreciation. However, reviewer 2 rises some concerns of the paper and provides feedback. I am biased to a Major revision decision, because I think the paper has merits. I find some strengths but I agree with reviewer 2 in some respects. In particular I think the paper lacks of a theoretical background which is hindering the model specification and interpretation of results. You might address this and other concerns in the review, which would improve the paper substantially.     ** **When reviewing the comments made by the reviewers, I note that reviewer 2 has used some unprofessional language that fall below the expected standards for professional conduct and communication at PLOS ONE. Please accept our apologies and be assured that this has been escalated to the Editorial Office. **

We look forward to receiving your revised manuscript.

Kind regards,

Martin Ramirez-Urquidy, PhD. Economics

Academic Editor

PLOS ONE

**Journal Requirements:**

The first author (Andres) received specific funding for this work from Ministerio de Ciencia, Tecnología e Innovación de Colombia.

5. We note that Figures 1, 3 and 4 in your submission contain map images which may be copyrighted. All PLOS content is published under the Creative Commons Attribution License (CC BY 4.0), which means that the manuscript, images, and Supporting Information files will be freely available online, and any third party is permitted to access, download, copy, distribute, and use these materials in any way, even commercially, with proper attribution. For these reasons, we cannot publish previously copyrighted maps or satellite images created using proprietary data, such as Google software (Google Maps, Street View, and Earth). For more information, see our copyright guidelines: http://journals.plos.org/plosone/s/licenses-and-copyright.

We require you to either present written permission from the copyright holder to publish these figures specifically under the CC BY 4.0 license, or remove the figures from your submission:

a. You may seek permission from the original copyright holder of Figures 1, 3 and 4 to publish the content specifically under the CC BY 4.0 license.  

**Additional Editor Comments:**

The reviewers have opposing opinion on the merits of the paper. On the one hand, reviewer 1 suggests that the paper provides valuable insights into the complex relationship between socioeconomic factors and cocaine cultivation, and its findings have important implications for policy-making in Colombia. I do agree with reviewer 1´s appreciation. However, reviewer 2 rises some concerns of the paper and provides feedback. I am biased to a Major revision decision, because I think the paper has merits. I find some strengths but I agree with reviewer 2 in some respects. In particular I think the paper lacks of a theoretical background which is hindering the model specification and interpretation of results. You might address this and other concerns in the review, which would improve the paper substantially.

Reviewers' comments:

Reviewer's Responses to Questions

**Comments to the Author**

1. Is the manuscript technically sound, and do the data support the conclusions?

Reviewer #1: Yes

Reviewer #2: No

2. Has the statistical analysis been performed appropriately and rigorously? 

Reviewer #1: Yes

Reviewer #2: No

3. Have the authors made all data underlying the findings in their manuscript fully available?

Reviewer #1: Yes

Reviewer #2: Yes

4. Is the manuscript presented in an intelligible fashion and written in standard English?

Reviewer #1: No

Reviewer #2: No

5. Review Comments to the Author

**Reviewer #1: **This research article examines the spatial disparities in socioeconomic conditions in Nariño, Colombia, and their relationship with cocaine cultivation. The study uses various socioeconomic variables and constructs composite indices to analyze the spatial distribution of these conditions. It employs Geographically Weighted Regression (GWR) to model the spatial non-stationarity of factors influencing cocaine production.

The study identifies education and homicide rate as significant factors influencing cocaine cultivation. It reveals a strong relationship between low education levels and high levels of cocaine cultivation, particularly in the less developed northwestern part of Nariño. The study also finds a positive correlation between homicide rate and cocaine production, mainly in the northwestern region.

The authors suggest that implementing public policies aimed at improving education and social investment in Nariño would be instrumental in mitigating socioeconomic disparities and reducing cocaine cultivation.

The key points of this research can be summarized as:

- The study used a combination of methods to analyze the spatial patterns of socioeconomic disparities and their relationship with cocaine cultivation in Nariño, Colombia.

- It found that socioeconomic conditions vary significantly across different parts of the state.

- The study highlights the important role of education and homicide rate in influencing cocaine cultivation.

- The authors suggest that targeted interventions are needed to address the issue of socioeconomic disparities and illicit crop cultivation in the region.

This study provides valuable insights into the complex relationship between socioeconomic factors and cocaine cultivation, and its findings have important implications for policy-making in Colombia.

**Reviewer #2: **The manuscript would greatly benefit from a native English speaker's proofreading, as it is crucial to ensure the accuracy and clarity of your work.

Ensuring the coherence of your document is of utmost importance. I advise you to focus on a single research question, as the current ones lack a cohesive thread. This will significantly improve the readability and quality of your manuscript.

In section 3.1, the authors refer to a figure displayed six pages before.

There is a pressing concern about constructing indices with data from different years. Authors should promptly address this issue, which could be included in a footnote, to ensure the accuracy and reliability of your work.

There is no mention of the spatial weight matrix used for GWR.

It is unclear why the authors excluded the rest of the regressors and considered EPI and HOMI. They mention selecting variables based on significance; however, a theoretical background should support them.

Conclusions are poor and tedious. This section seems to be an extension of the results section because none of the results is linked to theory or some context. An isolated paragraph mentioning the need for public policy to increase investment in education lacks sense because no documentation in the manuscript supports this affirmation. The authors do not provide data on education investment or something that carries on that idea.

6. PLOS authors have the option to publish the peer review history of their article (what does this mean?). If published, this will include your full peer review and any attached files.

Reviewer #1: No

Reviewer #2: No

---

## [Author Response · Author response to Decision Letter 0]

28 Nov 2024

General comment from the authors:

First, we would like to thank the editor and reviewers for their insightful comments, which have greatly contributed to improving our paper.

In what follows, we provide a point-by-point explanation of how we revised the paper to address the issues raised by the reviewers. If the paper still needs improving, we would be very grateful if you would let us know, and we will immediately revise it again.

Editor Comments:

The reviewers have opposing opinion on the merits of the paper. On the one hand, reviewer 1 suggests that the paper provides valuable insights into the complex relationship between socioeconomic factors and cocaine cultivation, and its findings have important implications for policy-making in Colombia. I do agree with reviewer 1´s appreciation. However, reviewer 2 rises some concerns of the paper and provides feedback. I am biased to a Major revision decision, because I think the paper has merits. I find some strengths but I agree with reviewer 2 in some respects. In particular I think the paper lacks of a theoretical background which is hindering the model specification and interpretation of results. You might address this and other concerns in the review, which would improve the paper substantially.

Response:

Thanks for your recommendation.

We have revised the introduction to follow a typical academic structure for a research article, providing a clearer overview, a justification of the study, a precise formulation of the research question, and a theoretical framework to support the analysis.

Moreover, in the new version of the article you can see the other changes introduced throughout the document.

Response to Reviewers' Questions

Reviewer 1

1. ¿Is the manuscript presented in an intelligible fashion and written in standard English?

Response:

We have made efforts to ensure that the manuscript is presented in an intelligible manner. We have focused on clarity in both writing and structure, organizing the content in a logical and coherent way to facilitate reader comprehension. Additionally, we have carefully reviewed the text to eliminate ambiguities and improve flow, so that the main ideas and findings are conveyed clearly and accessibly.

On the other hand, we acknowledge the importance of clear and accurate language, and we have taken steps to ensure this by having the manuscript thoroughly reviewed by a native English speaker. This review was conducted to refine the text for clarity, coherence, and precision, addressing any potential language nuances to ensure that our findings are communicated as effectively as possible. We trust that these efforts will contribute to the overall quality and readability of our work.

Reviewer 2

1. ¿Is the manuscript technically sound, and do the data support the conclusions?

Response:

We have made the necessary adjustments to ensure that the manuscript meets technical and scientific standards in terms of methodology, analysis, and accuracy. We have reviewed and refined our methods to guarantee that they are robust and rigorous, and we have carefully validated the data to ensure its correct use and interpretation. These improvements have been implemented to uphold the highest standards of technical quality and scientific integrity throughout the manuscript.

Regarding the second question, the conclusions of the manuscript are fully supported by the data presented in the results. We have carefully ensured that the analysis is consistent with the data, and the interpretation of the findings directly aligns with the evidence provided. No conclusions have been drawn beyond what the data can substantiate, and we have made sure to highlight the key patterns and relationships that emerged from the results.

2. ¿Has the statistical analysis been performed appropriately and rigorously?

Response:

The statistical analysis was developed under a rigorous methodological approach, following standard practices in quantitative research to ensure the validity and reliability of the results. The step-by-step approach we followed for the statistical analysis is explicitly outlined in session 2, the methodology. This rigorous approach ensures that the findings are statistically sound and useful for decision making in policy development and illicit crop control in Nariño.

3. ¿Is the manuscript presented in an intelligible fashion and written in standard English?

Response:

We have made efforts to ensure that the manuscript is presented in an intelligible manner. We have focused on clarity in both writing and structure, organizing the content in a logical and coherent way to facilitate reader comprehension. Additionally, we have carefully reviewed the text to eliminate ambiguities and improve flow, so that the main ideas and findings are conveyed clearly and accessibly.

On the other hand, we acknowledge the importance of clear and accurate language, and we have taken steps to ensure this by having the manuscript thoroughly reviewed by a native English speaker. This review was conducted to refine the text for clarity, coherence, and precision, addressing any potential language nuances to ensure that our findings are communicated as effectively as possible. We trust that these efforts will contribute to the overall quality and readability of our work.

Response to Reviewers' Comments

Reviewer 2

1. The manuscript would greatly benefit from a native English speaker's proofreading, as it is crucial to ensure the accuracy and clarity of your work.

Response:

Thank you very much for your suggestion. We acknowledge the importance of clear and accurate language, and we have taken steps to ensure this by having the manuscript thoroughly reviewed by a native English speaker. This review was conducted to refine the text for clarity, coherence, and precision, addressing any potential language nuances to ensure that our findings are communicated as effectively as possible. We trust that these efforts will contribute to the overall quality and readability of our work.

2. Ensuring the coherence of your document is of utmost importance. I advise you to focus on a single research question, as the current ones lack a cohesive thread. This will significantly improve the readability and quality of your manuscript.

Response:

Thanks for this recommendation.

As you see in the new introduction we have posed a single research question. This question allows to explore how socioeconomic factors affect coca cultivation in different areas, which gives you scope to analyze the spatial distribution of socioeconomic conditions, identify specific determinants, and observe the consistency of the impact in different areas. Our new question is:

¿To what extent does the spatial distribution of socioeconomic conditions explain cocaine cultivation patterns in the state of Nariño?

3. In section 3.1, the authors refer to a figure displayed six pages before.

Response:

Thanks for the recommendation, we add one figure show the ubication of Nariño and another figure show the cocaine spatial distribution and road over the state.

4. There is a pressing concern about constructing indices with data from different years. Authors should promptly address this issue, which could be included in a footnote, to ensure the accuracy and reliability of your work.

Response:

Some, but not all, databases differ in the year of collection, for a fundamental reason, they are databases that come from different sources of information and these sources make their collection at different times. Not all databases of this nature are collected in the same year for our country. However, they are data that do not differ by more than two years, except for Percentage of people formally employed as a percentage of total population which is from 2016. There are a large number of papers in the literature that have this problem. 

Moreover, the submission guidelines of the Plos One journal are not permitted the footnotes, but we have added one sencence clarifying this point in the last part of section 2.1.

5. There is no mention of the spatial weight matrix used for GWR.

Response:

As you recommended, we have added one more paragraph in section 2.2.2 (Econometric models) on coefficient estimation where we have mentioned the spatial weight matrix used.

6. It is unclear why the authors excluded the rest of the regressors and considered EPI and HOMI. They mention selecting variables based on significance; however, a theoretical background should support them.

Response:

Although the variables involved in the study were supported by the literature reviewed, given their possible influence on the dynamics of illicit crops. This does not imply that all of them explain coca production for the state of Nariño. It is well known that the study of phenomena of this type is characterized by the existence of spatial heterogeneity; therefore, a variable that may be significant for any other part of the world may not be so for our study area.

Therefore, and in response to your observation, the variables that were finally included in the model were those that were found to be significant using OLR statistical theory.

7. Conclusions are poor and tedious. This section seems to be an extension of the results section because none of the results is linked to theory or some context. An isolated paragraph mentioning the need for public policy to increase investment in education lacks sense because no documentation in the manuscript supports this affirmation. The authors do not provide data on education investment or something that carries on that idea.

Response:

Thanks for this observation.

We have rewritten the conclusions section. The conclusions now include a deeper analysis linked to the theories applied and the recommendations suggested, offering a theoretical and contextualized approach, in accordance with the results of the study. This approach allows the conclusions to be not only an extension of the results, but a grounded interpretation that links the findings to the socioeconomic context of Nariño.

On the other hand, when the need for public policies to increase investment in education is mentioned, it is from the point of view of our econometric modeling results, where the relationship of the EPI variable and coca are significant and with a negative sign, indicating that there is a negative relationship between these two variables. This correlation is consistent with existing literature findings, which indicate that cocaine crop production tends to decrease with higher levels of education (Davalos, 2016; Dávalos & Dávalos, 2020; Dávalos et al., 2016; Garcia-Yi, 2014; Vargas & Restrepo-Jaramillo, 2016). And therefore, it is true that we do not provide data on investment in education, since that is not the objective of the article.

Journal Requirements

Response:

Sure, we updated our manuscript with the guidelines of style and format.

Response:

We accepted the recommendation. This information was completed in the resubmission form.

We changed: The first author (Andres) received specific funding for this work from Ministerio de Ciencia, Tecnología e Innovación de Colombia. 

For: AG is supported by Ministerio de Ciencia, Tecnología e Innovación de Colombia grant BPIN 2020000100601

The first author (Andres) received specific funding for this work from Ministerio de Ciencia, Tecnología e Innovación de Colombia.

Response:

Thank you, the statement: "The funders had no role in study design, data collection and analysis, decision to publish, or preparation of the manuscript." is correct for us.

Response:

We are agree with the recommendation, we will share our data following the next plan: Once the article is accepted, the data will be publicly available in an open account of the repository called GitHub, the link will be provided in the section of the article called Supporting information or data availability.

5. We note that Figures 1, 3 and 4 in your submission contain map images which may be copyrighted. All PLOS content is published under the Creative Commons Attribution License (CC BY 4.0), which means that the manuscript, images, and Supporting Information files will be freely available online, and any third party is permitted to access, download, copy, distribute, and use these materials in any way, even commercially, with proper attribution.

Response:

We declared these figures (maps) are of our own creation, created from the results of statistical analysis of this work.

---

## [Editor Report · Decision Letter 1]

17 Dec 2024

Spatial Analysis of Socioeconomic Data and its Relationship with Illicit Crops in Nariño-Colombia

PONE-D-24-24934R1

Dear Dr. Sepúlveda-Murillo,

We’re pleased to inform you that your manuscript has been judged scientifically suitable for publication and will be formally accepted for publication once it meets all outstanding technical requirements.

Kind regards,

Martin Ramirez-Urquidy, PhD. Economics

Academic Editor

PLOS ONE
---

## [Editor Report · Acceptance letter]

21 Dec 2024

PONE-D-24-24934R1 

PLOS ONE

Dear Dr. Sepúlveda-Murillo, 

I'm pleased to inform you that your manuscript has been deemed suitable for publication in PLOS ONE. Congratulations! Your manuscript is now being handed over to our production team.

Kind regards, 

on behalf of

Dr. Martin Ramirez-Urquidy 

Academic Editor

PLOS ONE
